# On the Role of Computation in Reinforcement Learning

**Raj Ghugare** [1]   **Michał Bortkiewicz** [* 1 2]   **Alicja Ziarko** [* 1 3 4 5]   **Benjamin Eysenbach** [1]

## Abstract

How does the amount of compute available to a reinforcement learning (RL) policy affect its learning? Can policies using a fixed amount of parameters, still benefit from additional compute? The standard RL framework does not provide a language to answer these questions formally. Empirically, deep RL policies are often parameterized as neural networks with static architectures, conflating the amount of compute and the number of parameters. In this paper, we formalize compute bounded policies and prove that policies which use more compute can solve problems and generalize to longer-horizon tasks that are outside the scope of policies with less compute. Building on prior work in algorithmic learning and model-free planning, we propose a minimal architecture that can use a variable amount of compute. Our experiments complement our theory. On a set 31 different tasks spanning online and offline RL, we show that (1) this architecture achieves stronger performance simply by using more compute, and (2) stronger generalization on longer-horizon test tasks compared to standard feedforward networks or deep residual network using up to 5 times more parameters.[1]

## 1. Introduction

The standard view in reinforcement learning (RL) is to treat RL policies as static functions that map states to actions. Policies typically spend a fixed amount of compute, regardless of the complexity of making the correct decision in the underlying state. For example, for a humanoid robot, the action of where to move the right foot is easy, but figuring out how to efficiently pack furniture in a truck requires more deliberation.

In RL, this limitation is often solved by citing Moore's law (Moore, 1965) and simply scaling the number of policy parameters. But the compute required to make good decisions in different situations can span many orders of magnitude; using a static massive policy with a large number of parameters is wasteful. Additionally, as compute becomes cheap, the complexity of the world and sensory data also increase (Javed & Sutton, 2024).

*In this work we advocate for a computational view of RL, where computation time and parameter count (memory) are distinct axes. The first primary contribution of our paper is to take a step towards formalizing this view.* We formulate RL policies as bounded models of computation and prove that there exist MDPs where policies with more compute time are arbitrarily better than policies with less compute. Intuitively, policies with more compute time should learn more generalizing solutions, where static or computationally limited policies might instead learn heuristics which overfit their training (Zhou et al., 2023; 2024b). In this vein, we prove that there exist MDPs where policies with less compute can overfit on the training tasks and fail to generalize, whereas policies with more compute can generalize to longer-horizon tasks during evaluation (Myers et al., 2025).

*The second primary contribution of our work is to empirically show that RL policies which use more compute can achieve stronger performance as well as stronger generalization to longer-horizon unseen tasks.* Indeed, prior empirical work has also recognized the importance of using more computation to improve the performance of RL agents. Model-Based RL attempts this by learning a transition model and performing search using a planning algorithm (Sutton, 1991). Prior work in model-free planning uses various inductive biases like tree-search (Farquhar et al., 2018) or latent space planning (Oh et al., 2017; Silver et al., 2017) to train policies which generate solutions iteratively. Notably Guez et al. (2018); Tamar et al. (2016) have shown that using iterative architectures for RL policies with purely model-free RL objectives can display planning-like behavior (Bush et al., 2025). One limitation of these works is that the architectures proposed are restricted to tasks with a

---

[*]Equal contribution, author order decided via alphabetical order of the last name [1]Department of Computer Science, Princeton University [2]Warsaw University of Technology [3]University of Warsaw [4]Akces NCBR [5]Institute of Mathematics of the Polish Academy of Science. Correspondence to: Raj Ghugare <rg9360@princeton.edu>.

*Proceedings of the 43rd International Conference on Machine Learning*, Seoul, South Korea. PMLR 306, 2026. Copyright 2026 by the author(s).

[1]The code for our experiments is available on this website.

complete top down view. In our work, we propose a minimal recurrent architecture that can be trained on any task, to use a variable amount of compute with the same number of parameters (Figure 1). On over 30 tasks, we show that these recurrent policies improve significantly when using more recurrent steps. When trained to reach only nearby goals, we show that these policies outperform residual networks with 5 times more parameters when evaluated on unseen faraway goals. We also propose a way to estimate a numerical measure of the *value of compute*, analogous to the value of information studied in POMDPs.

Our work is reminiscent of recent work on length generalization in language models (Zhou et al., 2023; 2024b) that make use of iterative computation to promote this sort of generalization. Our experiments complement these results by empirically demonstrating that iterative computation can unlock horizon generalization in RL, and can do so without a special training dataset or curriculum (Cho et al., 2024; Lee et al., 2025). While prior work often attributes the success of iterative reasoning to the tasks that a model has seen before (Hanna & Corrado, 2025b; Lester et al., 2021), both our experiments and theory highlight that increasing compute (including via iterative function application) can improve performance on RL tasks in settings without language, pre-training, or multi-task training.

## 2. Related Work

**Model-based RL and planning.** There has been significant prior work in model-based RL on using additional compute to improve RL policies using planning (Schrittwieser et al., 2020; Hafner et al., 2019; Janner et al., 2021; Hansen et al., 2022; Heess et al., 2015). The model of the world (Sutton, 1991) is typically learned independent of the primary learning process. Previous works have proposed model-learning objectives that support the primary RL objective (Farahmand et al., 2017; Grimm et al., 2020; Voelcker et al., 2023) or learning policies and models using a single objective (Eysenbach et al., 2023a; Ghugare et al., 2023). Our work is more closely related to approaches that propose architectures that can learn to plan, without explicitly learning a model of the world (Tamar et al., 2016; Farquhar et al., 2018; Guez et al., 2019b; Oh et al., 2017; Guez et al., 2018). Guez et al. (2019a) show that planning-like computations can be learned using a convolutional LSTM architecture (Shi et al., 2015) and a purely model-free RL objective.

**Recurrent architectures for learning algorithmic solutions.** In the supervised learning literature, prior works have shown that recurrent networks (Rumelhart et al., 1986; Hochreiter & Schmidhuber, 1997; Graves et al., 2014; He et al., 2015) trained on outputs of an algorithm can learn

solutions that extrapolate to much longer inputs (Graves, 2017; Wang et al., 2025a; Jolicoeur-Martineau, 2025; Geiping et al., 2025; Schwarzschild et al., 2021; Bansal et al., 2022). In our experiments we also show that such an architecture learns policies that can solve more difficult tasks during evaluation. However, unlike the longer length evaluation in supervised learning, the horizon of evaluation tasks is longer in our RL setup. The architecture we use falls in the family of hybrid recurrent networks in which certain layers are repeated multiple times during the forward pass (Gers & Schmidhuber, 2001). Let $n_i, n_o$ be the input network and the output head respectively and $r$ be an intermediate network. Then the forward pass of a canonical recurrent architecture is of the form $n_o(r^k(n_i(x)))$. Many variants of such architectures have been proposed, mostly in the supervised learning literature, where $r$ could be a recurrent network (Gers & Schmidhuber, 2001; Geiping et al., 2025; Dehghani et al., 2019; Yang et al., 2024; Guez et al., 2019a), a residual network (Bansal et al., 2022; Schwarzschild et al., 2021), or a diffusion model (Du et al., 2024). In Section 5, we propose a new minimal variant of a recurrent network that worked well in our experiments. Recently, Jolicoeur-Martineau (2025); Wang et al. (2025a) have proposed multiple techniques like deep supervision, truncated gradients and early stopping (Graves, 2017) for training recurrent architectures in supervised learning tasks. We perform ablations (Section 6.7) to determine the benefits of these techniques for our RL experiments.

**Chain of thought reasoning in LLMs.** The performance of pre-trained language models can be improved significantly using inference time strategies without changing the parameters of the model (Brown et al., 2020; Wei et al., 2022b;a). Such strategies include prompting the model to "think step by step" (Wei et al., 2022b) or using self verification combined with a search algorithm to iteratively improve its output (Yao et al., 2023; Zhou et al., 2024a; Weng et al., 2023; Muennighoff et al., 2025). Our paper also shows that policies using more compute and the same number of parameters can significantly improve performance. But unlike pretrained LLMs, our work is focused on training RL policies from scratch.

## 3. Preliminaries

**Goal-Conditioned Reinforcement Learning.** The agent operates in a goal conditioned MDP $\mathcal{M}$ (Kaelbling, 1993), where $s_t \in \mathcal{S}$ are states, $a_t \in \mathcal{A}$ are actions and $g \in \mathcal{G}$ are goals. Before every episode, a start state and a goal is sampled from a task distribution $p(s_0, g)$. At every timestep, the agent samples an action using a goal-conditioned policy $\pi(a \mid s, g)$ and the next state is sampled using the transition dynamics $p(s_{t+1} \mid s_t, a_t)$. At each timestep the agent receives a reward $r_g(s, a)$ from the MDP. The objective of

a GCRL algorithm is to maximize the discounted sum of rewards:

$$J(\pi) = \mathbb{E}_{\substack{p(s_0,g),\pi(a_t,s_t,g),\\p(s_{t+1}|s_t,a_t)}} \left[ \sum_{t=0}^{H-1} \gamma^t r_g(s_t, a_t) \right] \quad (1)$$

Note that the task distribution could be different during evaluation $p_{\text{train}}(s_0, g) \neq p_{\text{test}}(s_0, g)$. The value function of a policy $v_\pi(s)$ is defined as $\mathbb{E}_{\pi,p} \left[ \sum_{t=0}^{H-1} \gamma^t r(s_t) \mid s_0 = s \right]$. When the reward is sparse (of the form $r_+$ upon reaching the goal and $r_-$ elsewhere) and the transitions are deterministic, we will use the term horizon of a task $(s, g)$ to mean the minimum number of timesteps that are needed to reach the goal $g$ from state $s$. Lastly, we also use deterministic policies which are special policies which output an action instead of a distribution, i.e., $a = \pi(s, g)$.

**Computational theory.** For our theoretical results, we will represent an RL policy using a Turing machine. A Turing machine $M$ is defined by its set of states, input and tape alphabets, a transition function, and an accept and reject state (Sipser, 2006). In our work, the input alphabet is always $\{0, 1\}$ and the input is a binary string $s$. We denote the length of this string as $|s|$. We use $w^K$ to denote the string that repeats $w$, $k$ times. $w^*$ denotes the set $\{w^k \text{ for } k \in \mathbb{N}\}$, and $\epsilon$ to denote the empty string. We only focus on single-tape deciding machines that take an input string $s \in \{0, 1\}^*$ and always transition to either the accept ($M(s) = 1$) or reject state ($M(s) = 0$) and only use a single-tape for their computation. We also use $\mathcal{O}(\cdot)$ and $o(\cdot)$ for the big-o and little-o notation respectively. Lastly, a function $t : \mathbb{N} \to \mathbb{N}$, where $t(n)$ is at least $\mathcal{O}(n \log n)$, is called time-constructible if the function that maps the string $1^n$ to the binary representation of $t(n)$ is computable in time $O(t(n))$ (Sipser, 2006). Time-constructibility is a technical property of time-functions, that is required for the proofs, we do not refer to it in our experiments. Please refer Sipser (2006) for additional details on computation theory, all of our proofs use the same terminology.

## 4. On the complexity of behaving optimally

In this section, we introduce a framework for thinking about the compute used by deterministic policies in the context of complexity theory (Sipser, 2006). For all the theoretical results, we assume that the MDP is deterministic and the action space is binary, noting that all discrete action MDPs can be reduced to another MDP with binary action spaces. We first introduce two definitions.

**Definition 1** (Time bounded Turing machines). (Hartmanis & Stearns, 1965) Let $M$ be a Turing machine that halts on all inputs $s \in \{0, 1\}^*$ and $t : \mathbb{N} \to \mathbb{N}$ be function. We say that $M$ is $t$-bounded if it decides all inputs $s$ in at most $t(|s|)$ steps.

**Definition 2** (Time bounded policies and policy classes). A deterministic policy $\pi(s)$ is said to be $t$-bounded if there exists a $t$-bounded Turing machine such that $\pi(s) = 1$ if $M(s)$ accepts and $\pi(s) = 0$ if $M(s)$ rejects, for all $s \in \{0, 1\}^*$. Here $s$ is the binary encoding of the state. We define a policy class $\Pi_t$, as the set of all deterministic policies that are $t$-bounded.

**Assumption 1.** For all policies we consider $\pi \in \Pi_t$, the description length of its corresponding Turing machine under a canonical representation scheme is always bounded by a constant, i.e., $|\langle M \rangle| < K_{max} \in \mathbb{N}$. $K_{max}$ can be as large as possible, but our theoretical results do require it to be no smaller than the length of a particular universal Turing machine.

In the real world this assumption is always true, as any feasible set of policies must rely on a finite amount of memory to store its parameters and code. Our main theoretical result is that policy classes with greater computational budgets are capable of solving a broader range of problems:

**Theorem 4.1** (Policy Hierarchy Theorem). *Let $g(n)$ and $t(n)$ be time-constructible functions such that $g(n) \in o(t(n)/\log t(n))$. Then there exists an MDP and a function $f(n) \in \mathcal{O}(t(n))$ such that:*

1. *The optimal policy $\pi^*$ belongs to the class $\Pi_f$.*

2. *For any policy $\pi \in \Pi_g$, $J(\pi)$ is arbitrarily lower than $J(\pi^*)$.*

This theorem tells us that there exist tasks on which policy classes that have more compute perform arbitrarily better than policies with less compute. See Appendix A for the proof, which uses the time hierarchy theorems (Sipser, 2006) and Assumption 1 to construct the desired MDP. Our experiments will empirically show that policies with additional compute can achieve higher returns in practice (Section 6). This result is interesting because it suggests that, under computational constraints, standard results about MDPs may no longer hold. For example, computation constraints may be reflected as partial observability, potentially explaining why prior experimental work has used non-Markov policies for solving "fully observed" tasks (Kapturowski et al., 2019; OpenAI et al., 2019; Petrenko et al., 2023). While recent work has argued that additional computation ("thinking" or "reasoning") is primarily useful because it enables policies to leverage multi-task pre-training (Hanna & Corrado, 2025a), Theorem 4.1 shows that the value of additional computation does not depend on multitask learning nor on pre-training.

Importantly, additional computation need not translate to larger hypotheses classes with weaker generalization. Rather, policies that use additional compute can provably exhibit stronger generalization. The intuition, which we

formalize below, is that a certain amount of compute capacity is required to represent the correct algorithm, and compute-constrained models will instead learn heuristics.

**Theorem 4.2** (Long Horizon Generalization). *Let $g(n)$ and $t(n)$ be time-constructible functions such that $g(n) \in o(t(n)/\log t(n))$. Then there exists a goal conditioned MDP and a function $f(n) \in \mathcal{O}(t(n))$ such that:*

1. *The optimal goal conditioned policy $\pi^*$ belongs to the class $\Pi_f$.*

2. *For all training tasks $(s_{train}, g_{train} \sim p_{train}(s, g))$, the best g-bounded policy $\pi_g^* \in \Pi_g$ has the same value function as $\pi^*$. But for infinitely many longer-horizon test tasks, $\pi_g^*$ is arbitrarily worse than $\pi^*$.*

The proof can be found in Appendix A. This theorem implies that a policy class with less compute can overfit on the training tasks of a more difficult problem and fail to generalize to longer-horizon tasks during evaluation. A simple example of this is language models being unable to solve GSM8K (Cobbe et al., 2021) problems in a single forward pass (constant compute), but solving them when provided with more compute using chain of thought (Wei et al., 2022b). In our experiments (Section 6.4), we show that RL policies using more compute generalize to longer-horizon tasks during evaluation.

# 5. A Minimal recurrent architecture for studying the benefits of computation

The primary aim of this paper is to theoretically characterize when and how recurrent computation can aid reinforcement learning (Section 4). To complement our theory, we propose a simple recurrent architecture that we will use for representing value functions and policies. This architecture is heavily inspired by components of the LSTM (Hochreiter & Schmidhuber, 1997) and GRU (Cho et al., 2014) architectures, though we have stripped away some components in attempts to make it as simple as possible while recapitulating the experimental phenomenon. This section will describe the architecture, and Section 6 will apply it in experiments.

**A Recurrent Block.** The key component of the architecture is a recurrent block which takes as input $x$ and the current hidden state $c_i$ and outputs the next hidden state, $c_{i+1}$. We will recursively apply this block $N$ times, so we can think about this computation as mapping an input $x$ to an output $c_n$. We initialize $c_0 = \vec{0}$. The recurrent block is composed of two linear layers, FORGET : $(x, c_i) \mapsto f_i$ and INPUT : $(x, c_i) \mapsto I_i$. The recurrent block uses these layers

in the following computation:

$$f_i = \sigma(\text{FORGET}_\theta([x, \tanh(c_i)])) \tag{2}$$

$$I_i = \tanh(\text{INPUT}_\theta([x, \tanh(c_i)])) \tag{3}$$

$$c_{i+1} = f_i \odot c_i + (1 - f_i) \odot I_i. \tag{4}$$

Because the new cell state is calculated as an interpolation between the previous cell state and the latest local computation, we call this layer an interpolation recurrent unit (IRU).

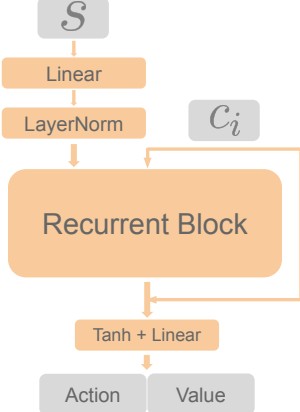

*Figure 1.* **Complete recurrent architecture.** This figure demonstrates the architecture for training policies/value functions. The recurrent block we use is an IRU.

**The Complete Architecture.** In our experiments, we will use this recurrent block as the core building block for our policy and value networks. These networks will take as input the observation and (for Q-functions) the action. After an initial linear layer and layer normalization layer (Ba et al., 2016), we apply the recurrent block layer $N$ times. After each application of the recurrent block, the previous cell state is added to the output using a skip connection. The final cell state $c_N$, after a Tanh activation, is passed through a final linear layer to predict the actions and values. The entire architecture is referred to as IRU-(N). See Fig. 1 for a visualization.

# 6. Experiments

The main goal of our experiments is to complement our theory and understand the impact of using additional recurrent steps on policy performance (Theorem 4.1) and generalization on longer-horizon tasks (Theorem 4.2). We evaluate the IRU architecture presented in Section 5 in both continuous and discrete settings. We describe the tasks and baselines in Section 6.1 and Section 6.2, and answer the following questions in the subsequent sections:

- Do more recurrent steps lead to better-performing policies? (Section 6.3)

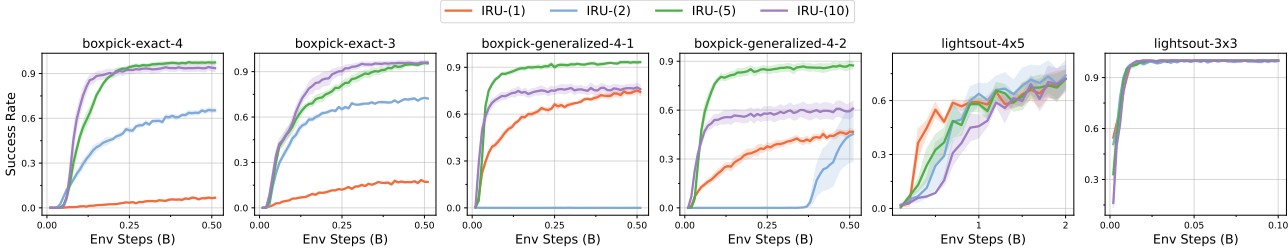

*Figure 2.* **Scaling recurrent steps in discrete environments.** Performance in most tasks improves as the number of recurrent steps increases, with performance often peaking at five recurrent steps.

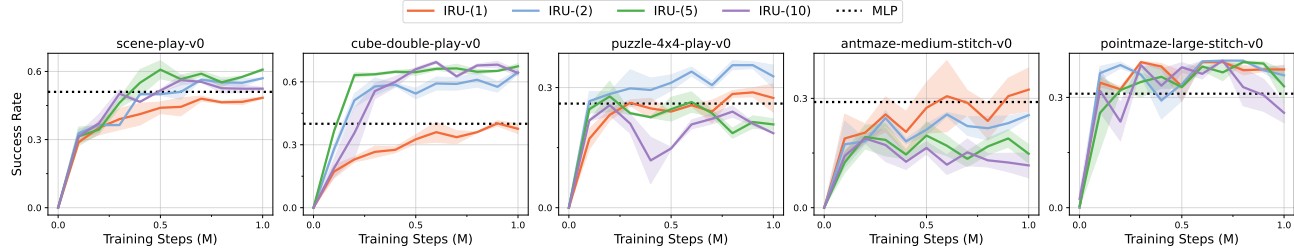

*Figure 3.* **Scaling up recurrent steps in continuous environments improves performance in OGBench tasks (Park et al., 2025).** Interestingly, additional recurrent steps considerably improve performance mainly in tasks that involve long-horizon reasoning (scene, cube, and puzzle), while performance in stitching navigation tasks increases marginally with more steps.

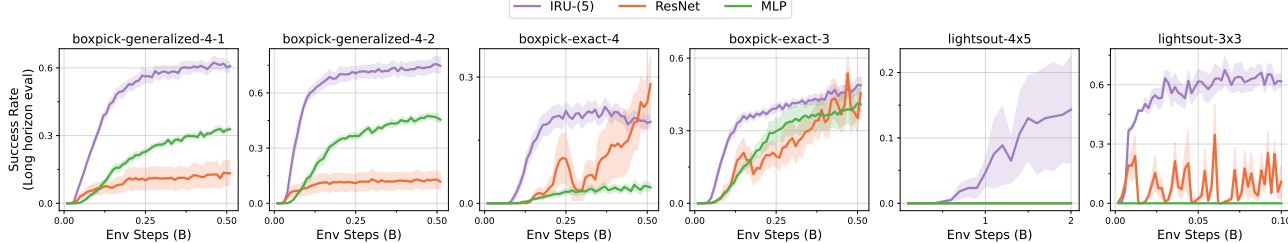

*Figure 4.* **Do recurrent steps improve generalization?** Throughout training, we track how policies learned by the IRU architecture and baselines perform on unseen tasks, including those that require more steps to solve. The IRU architecture learns faster (on $5/5$ tasks) and converges to higher asymptotic performance (on $4/5$ tasks) than MLPs and deep ResNets. The gains from IRU are most pronounced on `lightsout-4x4`, where all other architectures achieve only trivial performance.

- Does the recurrent architecture enable horizon generalization? (Section 6.4)

- How to measure the value of additional compute? (Section 6.5)

- Why does more compute improve performance? (Section 6.6)

- How do different recurrent blocks perform with additional compute? (Section 6.7)

### 6.1. Tasks

To evaluate the proposed architecture across a wide range of tasks — including discrete and continuous — we conduct experiments in the following domains: boxpick stitching benchmark (Bortkiewicz et al., 2025), lightsout puzzle (Anderson & Feil, 1998) and OGBench (Park et al., 2025).

**Boxpick.** The Boxpick benchmark consists of a discrete task on an $n \times n$ grid, where an agent navigates, picks up boxes, and places them on target locations. A rollout is successful if all boxes are placed on arbitrary targets, similar to Sokoban. We use two settings: *generalized stitching* (boxpick-gen-m−n) and *exact stitching* (boxpick-exact-m). In boxpick-gen-m−n, training tasks are sampled where only n boxes are not on target. During evaluation, longer-horizon tasks are sampled such that all the boxes are off the target. In the boxpick-exact-m, the agent is trained to move boxes between any two neighboring quadrants of the board, while evaluation is performed with boxes and targets on diagonally opposite quadrants to evaluate test-time generalization to longer-horizon tasks. These tasks are very difficult for RL algorithms, and many state of the art algorithms only achieve trivial performance on them (Bortkiewicz et al., 2025).

*Table 1.* **Sample efficiency and final performance of IRU vs. feed-forward architectures.** IRU achieves higher final performance across all tasks and better sample efficiency in most cases.

| Environment | Env Steps | IRU-(5) | ResNet | MLP |
| --- | --- | --- | --- | --- |
| boxpick-exact-4 | 50% | **94.8 ± 2.3** | 29 ± 15.2 | 25 ± 2.2 |
| | 100% | **97.4 ± 2.8** | 58 ± 10.7 | 37 ± 5.8 |
| boxpick-exact-3 | 50% | 82.6 ± 5.9 | **89.1 ± 3.1** | 51.6 ± 5.8 |
| | 100% | **95.5 ± 2.4** | 93.8 ± 4.4 | 67.0 ± 8.0 |
| boxpick-gen-4-1 | 50% | **91.4 ± 2.0** | 53.0 ± 8.6 | 64.4 ± 3.4 |
| | 100% | **93.3 ± 1.8** | 57.7 ± 8.1 | 73.2 ± 0.7 |
| boxpick-gen-4-2 | 50% | **84.8 ± 5.0** | 31.3 ± 6.2 | 56.8 ± 1.8 |
| | 100% | **87.4 ± 4.9** | 32.8 ± 5.4 | 64.3 ± 1.4 |
| lightsout-4x5 | 50% | **54.2 ± 7.7** | 1.0 ± 0.5 | 0.2 ± 0.2 |
| | 100% | **71.9 ± 8.9** | 0.2 ± 0.2 | 0.5 ± 0.2 |
| lightsout-3x3 | 50% | **100.0 ± 0.0** | 30 ± 5.7 | 2.3 ± 1.6 |
| | 100% | **100.0 ± 0.0** | 66 ± 13.6 | 1.5 ± 1.2 |

**Lightsout.** lightsout-m×n is a combinatorial puzzle where the state space is a m×n grid of black or white switches ($\{0,1\}^{mn}$). The action space contains $mn$ actions, each pushing a particular switch on the grid. Upon pushing, the colors of adjacent switches flip their color. At the start of every episode, a target grid configuration is sampled ($\{0,1\}^{mn}$). The agent receives a reward of -1 until it reaches this target. The space of possible inputs is very large ($2^{21}$ for lightsout-$4\times5$), hence it is challenging to memorize all the optimal actions. During training, we sample shorter horizon tasks and during evaluation we sample longer-horizon tasks.

**OGBench.** We use a total of five different environments consisting of a total of 25 tasks from OGBench. All tasks are offline goal reaching RL tasks where agents are supposed to learn using a fixed dataset. The success rate of agents is plotted throughout training. Antmaze-medium-stitch and pointmaze-large-stitch tasks test trajectory stitching, while scene-play, cube-double-play and puzzle-4x4-play test long-horizon reasoning and combinatorial generalization abilities.

### 6.2. Baselines

In all three settings, we use the same algorithm for all baselines with the standard hyper-parameters. The only change is the architecture used. We primarily compare with two baseline architectures – a standard feedforward network (**MLP**) and a residual network (**ResNet**). The MLP baseline contains a similar number of parameters to the IRU in all experiments. The ResNet baseline has the same effective depth as the IRU, i.e., it uses a similar amount of compute. But the number of parameters in ResNet scales with its depth. For instance, in the boxpick tasks the IRU contains 500K parameters, while the ResNet contains 3M parameters. These baselines help distinguish the effects of

amount of compute and the number of parameters used by RL agents.

Results are reported using five random seeds for each experiment. For additional details about the environments and the algorithmic implementation, please refer to Section B.

### 6.3. The effect of recurrent steps on policy performance

In both the discrete tasks (Figure 2) and the continuous tasks (Figure 3), we see that performance increases significantly with more recurrent steps. In the challenging boxpick tasks, or the manipulation tasks in ogbench, performance increases up to 8 times after increasing the recurrent steps from one to ten.

In Table 1, we compare IRU-(5), with the MLP and the ResNet architecture after 2.5 (50%) and 5 (50%) million environment steps. On all tasks, IRU-(5) outperforms the MLP, which uses less compute but similar number of parameters. Notably, IRU-(5) is able to solve the most challenging tasks like boxpick-exact-4, boxpick-gen-4-1 and lightsout-4x5, achieving a significant performance boost over the ResNet architecture. This is despite the ResNet using $\approx 2\times$ more compute and $\approx 5\times$ more parameters than IRU-(5). We hypothesize that this is due to the ResNet overfitting to seen tasks. This hypothesis is supported by the higher standard error values for the ResNet architecture in Table 1.

### 6.4. Generalization to longer-horizon tasks

We hypothesize that the solution learned by IRU policies will achieve better generalization to longer-horizon tasks during evaluation (Theorem 4.2). We also hypothesize that the recurrent inductive bias of IRUs will help them generalize better than ResNets, on tasks where the true solution requires iteratively applying some fixed computation (for e.g., lightsout). To empirically validate this hypothesis, we evaluate the performance of IRU, MLP and ResNet on a test distribution of unseen tasks that are strictly longer-horizon than the training tasks.

Figure 4 shows that IRU outperforms MLP on all tasks and ResNet on $4/6$ tasks, and the IRU is more sample efficient on all tasks. Importantly, IRU training is more stable than ResNet, which exhibits higher variance across seeds.

### 6.5. How to measure the value of compute?

Our theory proves that there are certain tasks which only policies with more compute can solve (Section 4) and previous experiments (Section 6.3 and Section 6.4) show that policies with more recurrent steps perform better. We now provide a method to estimate the **value of compute** for any given problem. Given two policies $\pi_{t_1}$ and $\pi_{t_2}$ which use $t_1, t_2$ recurrent steps respectively, the $n$-step value of

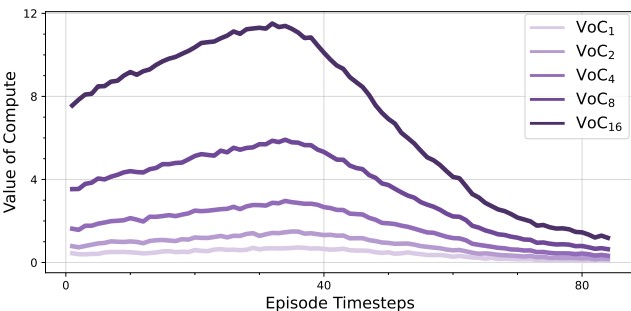

*Figure 5.* **The Value of Compute.** We plot the value of compute of IRU-$(1)$ over IRU-$(5)$ for different number of steps. We see that VoC increases with number of MDP time-steps using less compute. VoC also peaks at the early half of the episode where choosing the correct action is both difficult and crucial.

compute of $\pi_{t_2}$ over of $\pi_{t_1}$ is:[2]

$$\text{VoC}_n(s_0)$$
$$= v_{\pi_{t_1}}(s_0) - \left(\Sigma_{k=1}^n \gamma^{k-1} r(s_k, \pi_{t_2}(s_k)) + \gamma^k v_{\pi_{t_1}}(s_n)\right),$$

where $v_{\pi_t}$ is the value function of $\pi_t$. Intuitively, the value of compute measures how much value is gained/lost by temporarily using $t_2$ recurrent steps (instead of $t_1$) for $n$ time-steps in the MDP.

In Figure 5, we plot the value of compute for the boxpick-exact-4 with $t_1 = 5$ (IRU-$(5)$) and $t_2 = 1$ (IRU-$(1)$) over the course of several successful trajectories. We observe that value of compute increases with $n$, in line with the intuition that an agent will lose more value if it continues to use less compute for longer. For all values of $n$, the value of compute increases and peaks in the early half of the episode and then decreases towards zero. This is also expected since during the early half of an episode, selecting the correct action is both difficult and crucial. If the agent behaves sub-optimally, due to less compute, for too long, it might enter a region of the state space where it no longer knows how to recover. Towards the end of the episode when most blocks are already in their place, the value of compute is low as it is easier to select the correct action. This experiment highlights an opportunity for future work: compute-optimal agents should adaptively decide how much compute to use based on the difficulty of the current state.

### 6.6. Why does more compute improve performance?

While experiments show that using more compute markedly improves policy performance and the theory provides expressivity results, it remains unclear whether this improvement stems from stronger exploration (i.e., better data) or from applying a higher capacity model to those data. It

---

[2]For simplicity we assume deterministic transitions and policies. An expectation needs to be added to the second and third term for the general case.

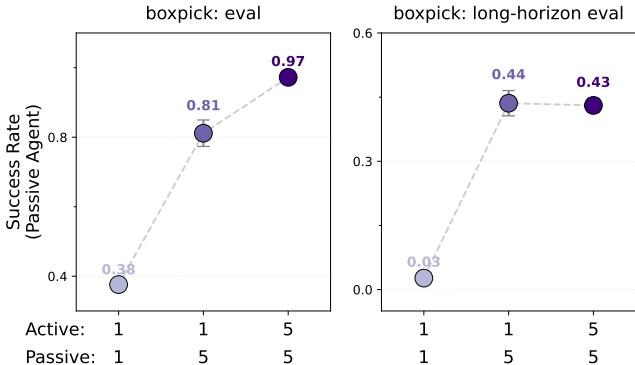

*Figure 6.* **Yoked experiments (Ostrovski et al., 2021) to understand the performance benefits of using more compute.** These results show performance of a passive agent that is trained using the data collected by an active agent. We see that the highest jump in performance comes when the passive agent uses five recurrent step even though the data collecting agent only uses one.

is important to dissect where the primary benefits of additional compute come from. We now propose an experiment (similar to (Ostrovski et al., 2021)) which allows us to do so.

Along the standard RL training loop, we train a passive agent on the data collected by original active agent. Both agents use the same objective and the architecture (but different weights), but the amount of recurrent steps they use are different. We perform this experiment on the boxpick-exact-4 and boxpick-gen-4-1 tasks and report the average success rates of the passive agent on training and testing distribution in Figure 6. Note that all reported algorithms like DQN (Mnih et al., 2013), CRL (Eysenbach et al., 2023b), and IQL (Kostrikov et al., 2021) achieved zero success rates with standard architectures in the long horizon evaluation of the boxpick-exact-4 task (Bortkiewicz et al., 2025).

In Figure 6, we see that the highest jump in performance comes when the passive agent uses $5$ recurrent steps. Remarkably, the data required to train the high-performing agent with $5$ recurrent steps are almost all contained in the data collected by an agent that used just $1$ recurrent step, even though that agent itself received low returns. We also run experiments where the active agent uses five recurrent steps and the passive agent uses one. We observe that the performance of this passive agent is worse than the performance of IRU-$(1)$. This could be because the off-policy bias of training passive agents or the inability of policies with less compute to learn good policies despite high quality data. Increasing the compute used by the active agent improves the performance marginally on the training tasks, but slightly reduces the performance on evaluation tasks. This suggests that most of the benefits of more compute comes from the additional expressivity of the agent. While we expect additional expressivity to indirectly improve ex-

ploration, using additional compute to explicitly improve exploration is an interesting direction for future work.

## 6.7. Ablation studies for architectural choices.

We perform an ablation experiment to compare the results when substituting three other choices of recurrent blocks in Figure 1, instead of the IRU. We use an LSTM network (Hochreiter & Schmidhuber, 1997), feedforward residual network (He et al., 2015), and the IRU with the deep recursion technique from Jolicoeur-Martineau (2025). For deep recursion, the network maintains two latent variables and performs two hierarchies of recurrence (see Figure 3 of Jolicoeur-Martineau (2025) for more details). All the networks use the same number of recurrent steps. Figure 7 shows that the LSTM marginally outperforms the IRU. We prefer the IRU because it is twice as fast as an LSTM and contains half as many parameters at the same depth and hidden state dimensions. We see that the recurrent ResNet blocks do not achieve the same benefits of added compute. This suggests that a suitable architecture (some gating mechanisms in our case) are necessary to achieve the scaling benefits of additional compute. We also observe that the deep recursion technique works relatively worse. We believe that there is significant room for improving architectures as well as techniques that efficiently leverage the benefits that additional compute can provide in RL.

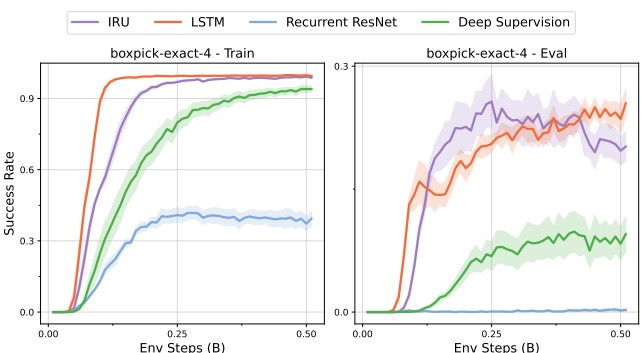

*Figure 7.* **Ablation experiments comparing several recurrent blocks in the complete architecture (Figure 1).**

## 7. Conclusion

In this work, we challenge the standard view of RL policies as static functions that map states to actions. Instead we advocate for a computational view of RL policies where computation time can be independent from the number of policy parameters. We formulate RL policies as bounded models of computation and prove that policies which use more compute can perform and generalize to longer-horizon evaluation tasks better than policies which use less compute. For our experimental results, we propose a minimal architecture that is trainable at different levels of compute and

show that using more compute can lead to significant gains in performance and longer-horizon generalization which was elusive for strong prior algorithms with standard architectures (Bortkiewicz et al., 2025). Finally, we also provide a computational method to estimate the value of compute in RL, akin to the value of information in partially observed tasks. These results help us measure, even in fully observed tasks, the value that additional compute provides.

While prior work has characterized the computational complexity of common neural network architectures (Siegelmann & Sontag, 1992; Strobl et al., 2024), the implications of such computational constraints on reinforcement learning remain underexplored. For instance, common knowledge that all MDPs have a deterministic policy solution is no longer true when using computationally bounded policies. In such cases, questions of how to tradeoff compute and value and how to leverage memory become exciting directions for future work.

One limitation of our work is that we use a fixed amount of recurrent computation steps for all states. We do not demonstrate how the same policy can use an adaptive amount of compute depending on the difficulty of the current state. Future work could explore such methods which might automatically learn inference-time compute scaling strategies or pre-fetch anticipated computation to store in memory. Additionally, our empirical evaluation focuses on a minimal recurrent architecture to isolate the effects of computation. We do not explore transformer-based architectures, and given their ubiquity in machine learning, an exploration of transformer-based recursive architectures in RL remains an interesting direction for future research. Lastly, our theoretical results use a single tape Turing machine as a computational model and only focus on time-complexity. Similar interesting results could be proven for space complexity or by using computational models like boolean circuits which better resemble modern neural networks.

**Acknowledgements.** The authors are pleased to acknowledge that the work reported on in this paper was substantially performed using the Princeton Research Computing resources at Princeton University which is consortium of groups led by the Princeton Institute for Computational Science and Engineering (PICSciE) and Office of Information Technology's Research Computing. This research was also supported by grants from NVIDIA and utilized NVIDIA A100-SXM4-80GB GPUs, National Science Centre, Poland (grant no. 2023/51/D/ST6/01609), and Polish high-performance computing infrastructure PLGrid (HPC Center: ACK Cyfronet AGH) grant no. PLG/2025/018637. We gratefully acknowledge Polish high-performance computing infrastructure PLGrid (HPC Center: ACK Cyfronet AGH) for providing computer facilities and support within computational grant no. PLG/2025/018581. AZ is partially funded by Google PhD Fellowship. The authors thank Brenden Lake, Mahsa Bastankhah and Abhishek Panigrahi for helpful discussions.

## Impact Statement

This paper presents work whose goal is to advance the field of machine learning. There are many potential societal consequences of our work, none of which we feel must be specifically highlighted here

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

# A. Proofs

**Theorem A.1** (Policy Hierarchy Theorem). *Let $g(n)$ and $t(n)$ be time-constructible functions such that $g(n) \in o(t(n)/\log t(n))$. Then there exists an MDP and a function $f(n) \in \mathcal{O}(t(n))$ such that:*

1. *The optimal policy $\pi^*$ belongs to the class $\Pi_f$.*

2. *For any policy $\pi \in \Pi_g$, $J(\pi)$ is arbitrarily lower than $J(\pi^*)$.*

*Proof.* Let $U$ be a universal Turing machine that given an input of the form $(\langle M \rangle 10^*)$, simulates the computation of $M(\langle M \rangle 10^*)$ such that each step of the simulating $M$ takes up some time $c(|\langle M \rangle|)$, depending only on the size of $M$ and not on its input[3]. See Theorem 9.10 of (Sipser, 2006) for the description of this Turing machine $U$.

We will briefly re-describe how $U$ works. $U$ has two tracks (odd positions and even positions) in its single working tape. One of the tracks contains the information on $M$'s tape, and the second contains $M$'s current state and a copy of $M$'s transition function. During the simulation, $U$ always keeps the information on the second track near the current position of $M$'s head on the first track. Every time $M$'s head moves, $U$ shifts all the information on the second track near the new head position. Because the information on the track depends only on $|M|$, shifting it and reading $M$'s next action for every simulation step of $M$ takes at most a constant amount of time depending on $|M|$.

We now define a language $L$ that it is decidable in $\mathcal{O}(t(n))$, but cannot be decided in $o(t(n)/\log t(n))$.

$$L = \{(\langle M \rangle 10^*) : U \text{ rejects } (\langle M \rangle 10^*)$$
$$\text{in } \leq \frac{t(|\langle M \rangle 10^*|)}{\log t(|\langle M \rangle 10^*|)} \text{ steps}\} \tag{5}$$

The proof for this can be found in Theorem 9.10 of Sipser (2006) (see Theorem 3.1 of Arora & Barak (2006) for the proof of a simpler case). The proof constructs a Turing machine $D$ that decides $L$ in $f(n) \in \mathcal{O}(t(n))$. The proof then shows that for any $g(n) \in o(t(n)/\log t(n))$ bounded Turing machine $M$, there exists a constant $n_0 \in \mathbb{N}$ such that on all inputs of the form $\{(\langle M \rangle 10^*) \in \{0,1\}^{n>n_0}\}$ it will output the opposite of $D$. That is if $D$ accepts, $M$ will reject and vice versa. Hence proving that any $g \in o(t(n)/\log t(n))$-bounded Turing machine cannot decide $L$. In Assumption 1, we need $K_{max}$ to be larger than $|D|$. This is not a stringent requirement as D follows a relatively simple algorithm whose pseudo-code can be described in seven english statements in Theorem 9.10 of (Sipser, 2006).

Even though Theorem 9.10 of (Sipser, 2006) shows that the constant $n_0$ exists for any $g$-bounded Turing machine, it can be arbitrarily large. We will use Assumption 1 to bound this constant. The Turing machine $U$ computes $M(\langle M \rangle 10^*)$ on inputs of the form $(\langle M \rangle 10^*)$. We know that each step of this simulation can be done in constant time $c(|\langle M \rangle|)$. If $M$ is a $g(n)$-bounded Turing machine then $U$ will simulate it in at most $c(|\langle M \rangle|)g(|(\langle M \rangle 10^*)|)$ steps. Because of Assumption 1, the maximum time that $U$ will take to simulate any $g$-bounded Turing machine will be $c(K_{max})g(|(\langle M \rangle 10^*)|)$. Since $g(n) \in o(t(n)/\log t(n))$, there exists an $n_{max}$ such that for all $n \geq n_{max}$, $g(n)c(K_{max}) < t(n)/\log t(n)$. Hence, the running time of any $g$-bounded Turing machine on inputs of the form $(\langle M \rangle 10^*)$ with length greater than $n_{max}$ is going to be less than $\frac{t(|\langle M \rangle 10^*|)}{\log t(|\langle M \rangle 10^*|)}$. Such strings will be in $L$ if $M$ rejects and won't be in $L$ if $M$ accepts. Because $D$ decides $L$, any $g$-bounded Turing machine $M$ will output opposite to $D$ on inputs of the form $(\langle M \rangle 10^*)$ with length greater than $n_{max}$.

Using this language, we will define an MDP which satisfies both conditions of Theorem A.1. Each episode in the MDP will end in a single time-step. The state space of this MDP is the following:

$$\mathcal{S} = \{(\langle M \rangle 10^*) : |\langle M \rangle| < K_{max}\} \tag{6}$$

The action space is binary and the reward function is:

$$R(s,a) = \begin{cases} 0 & \text{if } a = D(s) \\ -R & \text{otherwise} \end{cases} \tag{7}$$

---

[3]This does not mean that $U$ runs in constant time; the total deciding time still depends on the length of $w$.

The start state distribution $p(s_0)$ is:

$$\langle M \rangle \sim \mathcal{U}\big(\{0,1\}^{\leq K_{max}}\big) \text{ and } 10^q \sim p(10^{l > n_{max}}), \tag{8}$$

where $p$ can be any distribution.

Condition 1 of Theorem A.1 is true because we know that $D$ is a $f \in \mathcal{O}(t(n))$-time Turing machine. Hence, $\pi^*$ will just be the policy that is simulated by $D$.

Condition 2 of Theorem A.1 is true because for any $\pi \in \Pi_g$, there are infinitely many states of the form $s = (\langle M_\pi \rangle 10^{l > n_{max}})$, where $\pi$ will get arbitrarily low reward $(-R)$. Because such states occur with probability $\frac{1}{2^{K_{max}+1}-2}$, the maximum returns that any $g$-bounded policy can get is $\frac{-R}{2^{K_{max}+1}-2}$. This value can be reduced arbitrarily by reducing the value of $-R$. $\quad\square$

**Theorem A.2** (Long Horizon Generalization). *Let $g(n)$ and $t(n)$ be time-constructible functions such that $g(n) \in o(t(n)/\log t(n))$. Then there exists a goal conditioned MDP and a function $f(n) \in \mathcal{O}(t(n))$ such that:*

1. *The optimal goal conditioned policy $\pi^*$ belongs to the class $\Pi_f$.*

2. *For all training tasks $(s_{train}, g_{train} \sim p_{train}(s,g))$, the best $g$-bounded policy $\pi_g^* \in \Pi_g$ has the same value function as $\pi^*$. But for infinitely many longer-horizon test tasks, $\pi_g^*$ is arbitrarily worse than $\pi^*$.*

*Proof.* We will use a lot of the components built in the policy hierarchy theorem (Theorem A.1) in this proof.

We first define the MDP. The state and goal space of the MDP are:

$$\mathcal{S} = \{(\langle M \rangle 10^*) : |\langle M \rangle| < K_{max}\},$$
$$\mathcal{G} = \{10^*\} \cup \{\epsilon\}.$$

The action space is binary and the reward is:

$$R_g(s,a) = \begin{cases} +R & \text{if } s \text{ is of the form } (\langle M \rangle, g) \\ 0 & \text{otherwise.} \end{cases}$$

The transition dynamics of the MDP are the following:

$$T((\langle M \rangle, w), a) = \begin{cases} (\langle M \rangle, w_{1,\dots,|w|-1}) & \text{if } a = D(s) \text{ and } w \neq \epsilon \\ \text{Terminal State} & \text{otherwise.} \end{cases}$$

Hence, at every step, if the policy outputs the correct action, the last bit of the current state string $w$ is removed. If the policy outputs the wrong answer, then the episode will end with a return of 0.

The training task distribution is a uniform distribution over the following set:

$$\{(\langle M \rangle 10^*) : |(\langle M \rangle 10^*)| < K_{\text{train}} < n_{max}\} \times \{\epsilon\}.$$

So the agent has to provide the correct answer at all time steps, till the string $w$ is reduced to an empty string. The agent will then receive a reward of $+R$. Note that the horizon of training tasks is always bounded by $K_{\text{train}}$.

Condition 1 of Theorem A.2 is true because, $\pi^*$ will just be the policy that is simulated by $D$ and we know that $D$ is a $f \in \mathcal{O}(t(n))$-time Turing machine.

For Condition 2, we note that the set of training tasks $\mathcal{S}_{train}$ is a finite set. Hence it is a regular language that can always be decidable in $\mathcal{O}(1) \subset o(t(n)/\log t(n))$. One can run the policy corresponding to this $g$-bounded Turing machine (since it is constant time, it is also $g$-bounded) and achieve the optimal returns on all training tasks. Let this policy be $\pi_g^*$ and let $M_g^*$ be the corresponding Turing machine. From Theorem A.1, we know that the there are infinitely many longer-horizon test tasks of the form $s_{test} = (\langle M_{\pi_g^*} \rangle 10^*), g_{test} = \epsilon$, with $|\langle M_{\pi_g^*} \rangle 10^*| > n_{max}$, where $\pi_g^*$ will take the suboptimal action and get zero returns. $\quad\square$

# B. Implementation details

We present the implementation details of all the results presented in our paper. Unless stated otherwise, the algorithm specific hyper-parameters used for all architectures are the same as standard hyper-parameters and hence are not tuned to favor any architecture. Only the architecture relevant code is changed when comparing IRU, MLP or ResNets. The underlying algorithm for all the boxpick tasks is goal conditioned DQN (He et al., 2015), for lightsout is PPO, and for OGBench tasks is goal conditioned IQL (Kostrikov et al., 2021).

**Boxpick.** We use the standard hyper-parameters used by the DQN algorithm implemented in Bortkiewicz et al. (2025) benchmark listed in Table 2. We use a single IRU layer to make the recurrent block. All linear layers have a dimension of 256. The ResNets used in this task consist of two residual blocks, each with four hidden layers, following the architecture of Wang et al. (2025b). The dimensions of all layers in the residual blocks is 256. The MLPs used in this task contain 2 hidden linear layers of 256 dimensions.

*Table 2.* DQN hyperparameters in Boxpick benchmark.

| Hyperparameter | Value |
| --- | --- |
| num env steps | 500,000,000 |
| num updates | 1,000,000 |
| max replay size (per env instance) | 10,000 |
| min replay size | 1,000 |
| episode length | 100 |
| discount | 0.99 |
| number of parallel envs | 1024 |
| batch size | 256 |
| learning rate | 3e-4 |
| target_entropy | 1.1 |
| layer_norm | True |

**OGBench.** We use the standard hyper-parameters used by the IQL algorithm implemented in Park et al. (2025) benchmark. We use a single IRU layer to make the recurrent block. All linear layers have a dimension of $64$. The baseline MLP results are directly taken from Park et al. (2025).

**Lightsout.** We implement the PPO algorithm and use standard hyper-parameters listed in Table 3. We use stack two IRU layers to make the recurrent block. All linear layers have a dimension of $64$. The ResNets used in this task contain $10$ residual blocks with two hidden layers each. The dimensions of all layers in the residuals blocks is $64$. The MLPs used in this task contain $4$ hidden linear layers of $256$ dimension.

*Table 3.* PPO Hyperparameters in lightsout benchmark.

| Hyperparameter | Value |
| --- | --- |
| number of parallel envs | 2048 |
| discount factor | 0.99 |
| gae parameter | 0.95 |
| clipping parameter | 0.3 |
| entropy coefficient | 0.01 |
| learning rate | $1 \times 10^{-4}$ |
| rollout length | 160 |
| optimization epochs | 8 |
| mini-batches per rollout | 32 |
| advantage normalization | True |

