# OpenReview forum: "On the Role of Computation in Reinforcement Learning"
_ICML.cc/2026/Conference — ICML 2026 spotlight_

### Official Review · Reviewer_Cs9M · 2026-02-26

**Soundness:** 3
**Presentation:** 3
**Significance:** 3
**Originality:** 4
**Overall Recommendation:** 5
**Confidence:** 4

**Summary:**

This paper introduces a theoretical framework for analyzing the effect of compute on optimal policy learning in RL and presents an empirical evaluation.
The authors model RL policies as time-bounded Turing machines and show that, for some MDPs, a policy with more compute time can maximize the learning objective arbitrarily better than a policy with less compute.
They further extend this result to show that additional compute leads to improved long-horizon generalization.
To instantiate the theory, the paper introduces a minimal recurrent architecture, Interpolation Recurrent Unit (IRU), which applies a recurrent block iteratively for a fixed number of steps.
Experimental results in both discrete and continuous environments support the theoretical analysis: increasing compute improves performance and yields stronger generalization compared to the considered baselines.
The authors also empirically study the value of compute and analyze why additional compute leads to better performance, providing further evidence for their claims.

**Compliance With Llm Reviewing Policy:**

Affirmed.

**Final Justification:**

The rebuttal has reinforced my initial assessment. Overall, I think it is a good paper. I recommend accepting it.

**Key Questions For Authors:**

1. The theoretical framework is developed for deterministic MDPs with binary actions. I understand that the results trivially extend to discrete action spaces; however, is that the case for continuous actions? I would appreciate clarification from the authors on this point. In particular, since the experiments include MDPs with continuous actions, it would be helpful to briefly discuss how (or whether) the theory extends to this case in the main body of the paper.
2. The legend for Figure 4 does not seem to match the plots, the caption, or the discussion in the main text, i.e., MLP and ResNet do not appear in the legend. Is this a bug, or am I misunderstanding the figure?

**Limitations:**

yes

**Strengths And Weaknesses:**

This paper addresses an important and timely question.
To the best of my knowledge, the effect of compute resources on policy optimality in RL has not been formally studied before.
Given the recent interest in test-time compute and adaptive inference, this is a result that the community needed.
The proposed theoretical framework is well-motivated, and, to the best of my understanding, the assumptions and proofs are sound.
In addition, the experiments are thorough and span a broad range of settings, providing a comprehensive empirical analysis that meaningfully complements the theory.

The theoretical results are developed for discrete action spaces, while part of the empirical evaluation focuses on continuous control.
Although this does not invalidate the experiments, the gap between the theory and the continuous-action results could be discussed more explicitly.
Conceptually, the main result aligns with the intuition that more compute should help, and the paper presents it clearly and is generally easy to follow.
My main suggestion regarding presentation is to include a high-level figure summarizing the theoretical results and their implications; such a figure could make the core contributions more accessible to a broader audience.
Finally, the legend for Figure 4 appears to be buggy, i.e., no MLP or ResNet in the legend.

---

> ### Author Rebuttal · Authors · 2026-03-30
>
> We thank the reviewer for their insightful feedback and constructive comments. We also thank the reviewer for their positive remarks about our work. Below, we address the reviewer’s questions:
>
> > discrete vs continuous action spaces
>
> In all real world experiments, continuous variables have some finite precision. For example, our experiments in OGBench use 32 bit floating point precision. These finite precision continuous variables can be viewed as finite length binary strings, hence are effectively discrete. Although our theory does not account for measure theoretical aspects of continuous variables, for all practical purposes it still holds true with finite precision "continuous" action spaces.
>
> Real numbers can be approximated using finite binary strings of increasing lengths. But we leave a proper measure theoretic treatment to future work. We have added this discussion in the main body and the limitation paragraph of our paper.
>
> > the legend for Figure 4 appears to be buggy
>
> We thank the reviewer for pointing this out. We have corrected this error. The correct legend is Orange=IRU, Blue=ResNet & Green=MLP.
>
>
> > include a high-level figure summarizing the theoretical results and their implications
>
> We have prepared this figure and will include it in the final draft of our paper. This figure summarizes our theoretical results with a simple tabular MDP as an example. It then shows how the same policy can be represented as a table (high memory, low compute) and as a python function (constant memory, higher compute).
>
> **We appreciate the reviewer's acknowledgement of our contributions. We hope that our answers address all the reviewer's concerns.**

---

> > ### Author Rebuttal · Reviewer_Cs9M · 2026-04-01
> >
> > I thank the authors for their response and will maintain my positive score.

---

### Official Review · Reviewer_DFGA · 2026-03-04

**Soundness:** 1
**Presentation:** 3
**Significance:** 2
**Originality:** 3
**Overall Recommendation:** 5
**Confidence:** 3

**Summary:**

This paper seeks to understand how more computation leveraged *within a policy* (what you would refer to as inference-time computation) affects the performance of said policy. They explore a number of axes for this; first by theoretically justifying that a higher-compute policy is more performant, and then using some empirical justification to demonstrate that more compute = better policies. They compare a simple recurrent network, called the Interpolation Recurrent Unit, against MLP and ResNet policies in three suites of environments, exploring trainability (in-distribution performance/sample efficiency) as well as generalisation performance.

**Compliance With Llm Reviewing Policy:**

Affirmed.

**Final Justification:**

The rebuttal covered many of my experimental concerns. As such, I am increasing my score from 3->5, and increasing my confidence from 2->3, to reflect that I no longer hold empirical concerns. My only remaining hesitation is that there are some areas of poor clarity, but I am content with the new results.

**Key Questions For Authors:**

- Do these analyses strictly apply to goal-conditioned RL, or was that simply a convenient setting in which to explore this notion of out-of-distribution goal generalisation?
- Are computation time and parameter count distinct axes? In practice, using a neural network at least, parameter count is often strictly linked to computation time. I think I appreciate what is being said, but it's not very clear. It seems that the experiments really show that repeatedly applying a small network, with a certain structure (and the inductive biases built into the architecture) are better than training a computationally equivalent, larger ResNet?
- Given the links to COT reasoning that are brought up in this paper, and the contraast (these experiments focus on tabula rasa RL), do you think these findings *empirically* would hold up when RL finetuning a pre-trained model?
- Do the findings in Figures 2,3,4,5,6 hold up when hyperparameters are tuned for the individual methods.
- To check my understanding - the activation to IRU is reset every time an action is taken right? It is a markovian policy?


Typos/Errors:
- In Figure 4, the legend is presumably meant to be showingan MLP/ResNet/IRU? Similarly, in the caption, should it be lightsou-4x5?
- In Figure 3, antmaze, why do the different runs continue for different lengths of time.

**Limitations:**

The authors dedicate some of their conclusion to limitations, which is appreciated. i think discussing the limited hyperparameter tuning is also necessary.

**Strengths And Weaknesses:**

As a high level summary:
I found this a reasonably well written paper. I think the idea of the analysis is good, and well-motivated. That said, I think some of the empirical experiments are lacking in some regards. While I believe this is a sufficient enough limitation to recommend weak rejection, I am potentially open to discussing these perceived limitations with the authors to potentially change this score.

Strengths:

- The paper is, for the most part, well written (see below).
- The analysis is well motivated. Figuring out how to optimally distribute policy compute is a fundamentally important question, and one which this paper begins to question.
- Using Turing Machines as a means for this analysis is well-motivated and interesting conceptually
- Grounding the theorems in examples is useful for helping non-theoretical people (such as myself) understand them.
- The proposals and discussions are all reasonable and intuitive. While maybe somewhat obvious, formalising the notion of inference-time compute and theoretically proving that it is beneficial is a clear benefit to the paper.
- In some scenarious, the conclusion does follow; increasing computation does improve performance. That said, please see discussion below.


Weaknesses:
- Despite saying that most of the writing is clear, I think the introduction could do better to concretely spell out what we're talking about with regards to computation. Motivating through MBRL -> MFRL -> recurrence is logical, but I think just saying that the recurrent policy can be applied multiple times before an action is taken (my understanding) would be good. I also found the theorems a bit hard to parse, principally because the introduction of Turing Machines is abridged and refers to prior text. Given there was almost half a page of room, expanding this would be good. Finally, I think lines 79:83 suffers from trying to be *too brief* to the extent that the menaing of the discussion about related work is lost.
- Some of the notation/what feels like methodological discussion (107:) feels like it's bled into the related work.
- While I appreciate using a slimmed down architecture, for analysis, it would be interesting to see whether the finding about inference-time computation also scales to other more conventional architectures (e.g., applying an LSTM/GRU multiple times) - rather than just comparing IRU to them in figure 7 as an ablation on a single environment.
- It feels that more concrete findings are being drawn than the results justify. The general story of the paper is that increasing computation will improve performance, but IRU(5) seems to regularly outperform IRU(10) in discrete environments, and in continuous environments the story is significantly less consistent.
- Furthermore, in fig 4 (I'm assumeing this is meant to be Blue=ResNet, Green=MLP, Orange=IRU) - there are 6 tasks, not 5, and the IRU performs worst in the first. That would be ok, but I'm slightly concerned about the differences between these methods when using the same hyperparameters. The ResNet, for example, has 6 times the parameter count yet uses identical hparams (e.g., learning rate) for training. I think, when making claims as strong as the paper about one methodology being > than another, it is necessary to give each a fair comparison - in RL, this means ensuring some degree of hyperparameter tuning for each method. It is worth noting, the hyperparameters are quoted as 'default', but given they differ between the environments it suggests that the original environment authors will have done some degree of hyperparameter tuning with some architectural setup. To me, as a more empirically-minded researcher, this feels like a significant glaring oversight.
- I found both the discussion and analysis of Value of Compute confusing.

---

> ### Author Rebuttal · Authors · 2026-03-30
>
> We thank the reviewer for their insightful feedback and constructive comments. It seems that the reviewer’s main feedback is regarding (1) hyperparameter ablations for all methods, (2) inconsistent performance of IRU(10) over IRU(<=5), (3) more experiments with conventional architectures (e.g. LSTMs).
>
> To address (1), we ran HP ablations for ResNets and IRU, and reran experiments with the best HPs.
>
> To address (2), we ran new experiments with better HPs; IRU(10) indeed shows consistent improvement over IRU(<=5).
>
> To address (3), we have run preliminary experiments with LSTMs showing similar trends of performance vs compute.
>
> **Do these answers, revisions and experiments address all the reviewer's concerns?**
>
> > ensuring some degree of hyperparameter tuning
>
> Below, we tuned the learning rates and then the batch size for both ResNet and IRU(5) on the boxpick-exact-4 environment.
>
> ResNet:
>
> | Learning Rate | 3e-5 | 1e-4 | 3e-4 | 1e-3 |
> | --- | --- | --- | --- | --- |
> | Success Rate | 0.43 ± 0.02 | **0.91 ± 0.03** | 0.58 ± 0.1 | 0.62 ± 0.04 |
>
> | Batch Size (LR 1e-4) | 256 | 512 | 1024 |
> | --- | --- | --- | --- |
> | Success Rate | 0.91 ± 0.04 | 0.91 ± 0.1 | **0.93 ± 0.01** |
>
> IRU:
>
> | Learning Rate | 3e-5 | 1e-4 | 3e-4 | 1e-3 |
> | --- | --- | --- | --- | --- |
> | Success Rate | 0.89 ± 0.03 | 0.96 ± 0.01 | **0.97 ± 0.02** | 0.86 ± 0.02 |
>
> | Batch Size (LR 3e-4) | 256 | 512 | 1024 |
> | --- | --- | --- | --- |
> | Success Rate | 0.97 ± 0.02 | 0.991 ± 0.003 | **0.993 ± 0.004** |
>
> **Best hyperparameters found:**
>
> **Resnet: 1e-4, 1024**
>
> **IRU: 3e-4, 1024**
>
> > IRU vs ResNets with tuned HPs
>
> Using the tuned HPs, we ran new experiments on 3 boxpick environments, exact-4, generalized-4-1, and generalized-6-1. Note from prior work [1], many SOTA algorithms struggle in these environments.
>
> | Model | Generalized 4-1 | Generalized 6-1 | Exact |
> | --- | --- | --- | --- |
> | IRU (10) | **0.85 ± 0.01** | **0.7 ± 0.07** | **0.42 ± 0.12** |
> | ResNet | 0.82 ± 0.07 | 0.22 ± 0.27 | 0.229 ± 0.27 |
>
> The above results with tuned HPs add further evidence to our hypothesis that “the recurrent inductive bias of IRUs will help them generalize better than ResNets” (see section 6.4)
>
> > Performance vs Compute with tuned HPs
>
> Using the tuned HPs, we compare IRU with 1,2,5, and 10 iterative steps. The results below show a consistent improvement in performance with more iterative steps. One hypothesis is that better HPs reduce the optimization challenges faced with more iterative steps.
>
> | Iterative Steps | 1 | 2 | 5 | 10 |
> | --- | --- | --- | --- | --- |
> | Generalized 4-1 | 0.06 ± 0.007 | 0.37 ± 0.13 | 0.78 ± 0.11 | **0.85 ± 0.01** |
> | Generalized 6-1 | 0.01 ± 0.01 | 0.0 ± 0.0 | 0.0 ± 0.0 | **0.7 ± 0.07** |
> | Exact | 0.0 ± 0.0 | 0.24 ± 0.007 | **0.43 ± 0.18** | 0.42 ± 0.12 |
>
> We are re-running all experiments with the latest HPs and will update the paper accordingly.
>
> > LSTMs with more compute
>
> We compare LSTMs with different iterative steps. We see that more compute does lead to improved performance, LSTM (1) always has the least performance despite using the same parameters. But the trend with more compute is not always consistent. We hope these results serves as a proof of concept and future work builds on finding better techniques to scale more complex models with compute.
>
> | Iterative Steps | 1 | 2 | 5 | 10 |
> | --- | --- | --- | --- | --- |
> | Generalized 4-1 | 0.18 ± 0.04 | **0.95 ± 0.02** | 0.84 ± 0.008 | 0.85 ± 0.07 |
> | Exact | 0.0 ± 0.0 | 0.41 ± 0.009 | **0.67 ± 0.26** | 0.49 ± 0.13 |
>
> > analyses strictly apply to goal-conditioned RL
>
> Theorem 1 does not use goal conditioned MDPs, so it applies to all MDPs. In theorem 2, goal conditioning was a convenient way to define long horizon generalization. But the results apply to all multi task MDPs, as a goal conditioned MDP is just a special multi task MDP. We have added a note to clarify this in our paper.
>
> > tuned HPs
>
> Figure 3 baselines were already tuned. We have updated Fig 5 and Fig 6 with better HPs for IRU. As the underlying phenomena of performance improvement with more compute steps hasn't changed (in fact it gets stronger), we see that the results of these didactic experiments also remain the same.
>
> > writing suggestions about related work and introduction & errors in figure
>
> We have incorporated all writing suggestions, clarified the Value of Compute section, and fixed the noted typos in Figure 4
>
> > legend Fig 4
>
> We have corrected this error. Orange=IRU, Blue=Resnet & Green=MLP.
>
> > is IRU markovian
>
> Yes.
>
> > connections to CoT and RL finetuning
>
> Yes, this has already been seen in the recent “deep think” LLMs which are finetuned via RL to use many iterations (tokens) before outputting the answer.
>
> **We appreciate the reviewer's acknowledgement of our theoretical contributions. Our experiments are meant to complement the theory. We hope that above new experiments with tuned HPs strengthen our empirical claims.**
>
> [1] Is Temporal Difference Learning the Gold Standard for Stitching in RL?

---

> > ### Author Rebuttal · Reviewer_DFGA · 2026-04-02
> >
> > I don't have much to say - besdies some of my writing concerns, and the fact that perhaps the LSTM experiments don't exhibit the same trend as strongly (htey still exhibit the trend, and there is something to be said for the IRU providing a more surgical tool for analysis), I ma pretty content that my concerns have been addressed.
> >
> > Score 3->5 to reflect this, as I no longer think this paper sits in a "borderline" camp.

---

### Official Review · Reviewer_jUJd · 2026-03-09

**Soundness:** 3
**Presentation:** 3
**Significance:** 3
**Originality:** 3
**Overall Recommendation:** 5
**Confidence:** 3

**Summary:**

The proposed paper argues that in RL computation time and parameter count should be treated as two separate axes.
The key claim is that giving a policy more compute (via recurrent steps) improves both performance and generalization to longer-horizon tasks, without needing more parameters. The authors formalize this with complexity theory (time-bounded Turing machines) and propose the IRU architecture to validate it empirically across 31 tasks.

**Compliance With Llm Reviewing Policy:**

Affirmed.

**Final Justification:**

I am maintaining my positive score as the paper provides a compelling framing on computation, with relevant insights and analysis.

**Key Questions For Authors:**

**Questions / Recommendations:**

- **Non-monotonic performance:** The authors should provide some intuition for why there is a sweet spot rather than monotonic improvement, and some practical guidance on how to choose the number of recurrent steps for a new task.

- **Missing ResNet baseline at matched parameter count:** The paper should test a ResNet with the same amount of parameters as IRU. This matters because overparameterized networks can sometimes perform worse due to overfitting or optimization difficulties.

- **Stochastic environments:** OGBench includes stochastic environments (e.g., teleport maze variants). Can you report performance in these environments to see whether the deterministic assumption is only a theoretical limitation, not in practice? The author experiments on the full OG-Bench suite, so adding the full table of all the experimental results in Appendix will be valuable for the paper.

- **Inference and training time cost:** Since a key selling point is that IRU improves performance without adding parameters, readers need to know the actual wall-clock cost of scaling recurrent steps.

- **Vanishing gradients:** The gradient stability should discussed. It would also be useful to know whether gradient issues become a limiting factor when scaling to more than 10 recurrent steps, which could partly explain the non-monotonic performance observed in some environments.

**Limitations:**

yes

**Strengths And Weaknesses:**

**Strengths:**

- Separating compute time from parameter count is a clean and underexplored framing in RL.

- The empirical setup is methodologically sound: the algorithm and hyperparameters are held fixed across all architectures, so differences are attributable to the architecture alone.

- The yoked experiment in Section 6.6 is interesting. A passive agent with 5 recurrent steps, trained only on data collected by a weak 1-step agent, still dramatically outperforms that agent. This strongly suggests the main benefit of more compute is expressivity rather than better exploration or data quality. This is a meaningful mechanistic insight.

- Results are tested across a diverse set of environments (discrete, continuous, online, offline).

**Weaknesses:**

- **Non-monotonic performance:** Performance peaks around 5 recurrent steps and does not improve monotonically, for example, IRU-10 sometimes underperforms IRU-5. This is notable and unexplained.

- **Missing ResNet baseline at matched parameter count:** The ResNet baseline uses roughly 5x more parameters than IRU to match compute depth. However, the paper never tests a ResNet with the same ~500K parameters as IRU.

- **Stochastic environments:** All theoretical results assume deterministic transition dynamics.

- **Inference and training time cost:** The paper does not report the computational overhead of using more recurrent steps at inference and training time. The authors note IRU is faster than LSTM, but never report how training and inference time scale with the number of recurrent steps, which is the practically relevant quantity.

- **Vanishing gradients:** With many recurrent steps, vanishing or exploding gradients are a well-known concern. The paper does not discuss whether this is a problem in practice for IRU, and if so, how it is handled. The interpolation mechanism in the IRU (Equation 4) is reminiscent of LSTM gating and may naturally mitigate this, but this should be explicitly verified.

---

> ### Author Rebuttal · Authors · 2026-03-30
>
> We thank the reviewer for their insightful feedback and constructive comments. We also thank the reviewer for their positive remarks about our work. It seems that the reviewer’s main feedback is regarding (1) non monotonic performance increase with compute steps, (2) smaller ResNet baseline, (3) experiments with stochastic MDPs, and (4) analysis of vanishing gradients.
>
>
> To address (1), to lift potential optimization issues, we ran new experiments with better HPs; we see that IRU(10) indeed shows consistent improvement over IRU(<=5).
>
> To address (2), we have run new experiments with a ResNet baseline with equal number of parameters as IRU.
>
> To address (3), we ran experiments on stochastic environments in OGBench.
>
> To address (4), we ran a new experiment to analyse the vanishing gradients of RNNs, LSTMs & IRUs.
>
> **Do these answers, revisions and experiments address all the reviewer's concerns?**
>
> > Inconsistent performance
>
> Following Reviewer DFGA's suggestion, we tuned learning rates (ResNet: 1e-4; IRU: 3e-4) and batch size (1024) on boxpick exact. Using these HPs, the performance of IRU does improve monotonically:
>
> | Iterative Steps | 1 | 2 | 5 | 10 |
> | --- | --- | --- | --- | --- |
> | Generalized 4-1 | 0.06 ± 0.007 | 0.37 ± 0.13 | 0.78 ± 0.11 | **0.85 ± 0.01** |
> | Generalized 6-1 | 0.01 ± 0.01 | 0.0 ± 0.0 | 0.0 ± 0.0 | **0.7 ± 0.07** |
> | Exact | 0.0 ± 0.0 | 0.24 ± 0.007 | **0.43 ± 0.18** | 0.42 ± 0.12 |
>
> One hypothesis is that better HPs reduce the optimization challenges faced by using many more iterative steps. Many SOTA RL algorithms struggle in above environments [1].
>
> >  ResNet with the same ~500K parameters
>
> Using tuned HPs, we ran new experiments with a smaller ResNet, where each block contained 2 hidden layers with 128 dimensions (~560K parameters). The results below compare this small ResNet with the larger ResNet and IRU(10):
>
> | Method | generalized 4-1 | generalized 6-1 | exact |
> | --- | --- | --- | --- |
> | IRU (10) | **0.85 ± 0.01** | **0.7 ± 0.07** | **0.42 ± 0.12** |
> | Resnet | 0.82 ± 0.07 | 0.22 ± 0.27 | 0.229 ± 0.27 |
> | Resnet small | 0.07 ± 0.05 | 0.00 ± 0.00 | 0.03 ± 0 |
>
> These results strengthen our empirical claims and underscore the importance of compute for policy performance.
>
> > Stochastic envs
>
> We performed experiments on the teleport environments with IRU-(1) and IRU-(2).
>
> | Method | antmaze-teleport-navigate-v0 | antmaze-teleport-stitch-v0 |
> | --- | --- | --- |
> | IRU (1) | 0.34 ± 0.02 | 0.17 ± 0.03 |
> | IRU (2) | **0.35 ± 0.09** | **0.192 ± 0.03** |
> | MLP | **0.35 ± 0.05** | **0.17 ± 0.05** |
>
> Similar to our paper, we do not see a significant improvement in performance in these navigation settings with increased compute. Hence, we conclude that the value of compute (see section 6.5) is not very significant in navigation. This makes sense because 2D navigation requires the agent to store / compress a 2D map. This can be efficiently compressed using euclidean geometry without needing a lot of computation.
>
> We have updated the paper with the full table of OGbench results in the main paper.
>
> > Inference and training time cost
>
> We looked at the average inference and training time for all the models we tried on the boxpick generalized 4-1 task. For inference, we look at a batch size of 1 and for training a batch size of 1024. The results below show that the inference time for IRU(10) is about half that of the deep ResNet. While both IRU and ResNet have similar training time. Infact the inference time of the IRU(10), which is dominated by the python to XLA dispatch time, is comparable to that of the MLP.
>
> | Model | Single batch Inference (ms) | Train batch (ms) |
> | --- | --- | --- |
> | MLP | 0.5028 ± 0.8327 | 1.7866 ± 1.0327 |
> | ResNet | 1.3396 ± 1.1906 | 4.5833 ± 1.4296 |
> | IRU(1) | 0.2865 ± 0.4069 | 1.7795 ± 0.5779 |
> | IRU(5) | 0.3192 ± 0.1705 | 4.4713 ± 1.0699 |
> | IRU(10) | 0.6001 ± 0.5295 | 7.4494 ± 0.9058 |
> | LSTM(1) | 0.2524 ± 0.1185 | 2.5822 ± 0.8037 |
> | LSTM(10) | 0.7089 ± 0.5510 | 12.4636 ± 0.9344 |
>
> > Vanishing gradients
>
> We perform a didactic experiment on computing the maximal depth of a parenthesis sequence, e.g., maximal depth of (()(())) is 3. All models have a similar number of parameters, and the sequence length for training / testing is 200. Below we look at the gradients of the loss in log scale at the final timestep (t=199) at different points in time:
>
> ### Log scale gradients
>
> | Timestep | RNN | LSTM | IRU |
> |---|---|---|---|
> | 0 | -inf | -0.63 ± 1.13 | -0.67 ± 1.13 |
> | 50 | -inf | -0.77 ± 1.48 | -0.69 ± 1.24 |
> | 100 | -inf | -0.75 ± 1.47 | -0.59 ± 1.16 |
> | 150 | -15.71 ± 16.19 | -0.77 ± -1.53 | -0.66 ± 1.18 |
> | 199 | 0.44 ± 0.03 | -0.69 ± 1.20 | -0.58 ± 1.15 |
>
> These results show that while the standard RNN suffers from vanishing gradients, both the IRU and LSTM are able to sustain healthy gradients until 200 timesteps in the past.
>
> [1] Is Temporal Difference Learning the Gold Standard for Stitching in RL?

---

> > ### Author Rebuttal · Reviewer_jUJd · 2026-04-04
> >
> > Thanks for the clarification, which solves my concerns. I'm looking forward to seeing the updated version of the manuscript. I will keep my positive score.

---

### Official Review · Reviewer_LiRX · 2026-03-24

**Soundness:** 3
**Presentation:** 3
**Significance:** 3
**Originality:** 3
**Overall Recommendation:** 4
**Confidence:** 2

**Summary:**

The paper uses time-bounded Turing machines to prove that policies with larger computational budgets can solve MDPs and generalize to longer-horizon tasks that are strictly impossible for compute-constrained policies. Empirically, they propose the Interpolation Recurrent Unit (IRU), a minimal architecture that allows an agent to use a variable number of recurrent computation steps without increasing the number of parameters. Across 31 distinct tasks spanning discrete and continuous domains, they demonstrate that increasing the number of recurrent compute steps leads to significantly better performance and stronger generalization to unseen, long-horizon tasks, outperforming standard MLPs and ResNets with up to 5 times more parameters.

**Compliance With Llm Reviewing Policy:**

Affirmed.

**Final Justification:**

The concerns were resolved, and I raised scores reflecting the comments.

**Key Questions For Authors:**

- Running IRU-(10) requires 10 recurrent forward passes per environment step. How is the practical trade-off between this increased wall-clock inference latency and the performance gains, particularly for real-time control scenarios?
- Theorems 4.1 and 4.2 are strictly grounded in discrete Turing machines with binary inputs/actions. How do you intuitively bridge these theoretical guarantees to the continuous state and action spaces,

**Limitations:**

yes

**Strengths And Weaknesses:**

# Strength
- By comparing the IRU to parameter-matched MLPs and depth-matched ResNets, they isolate the specific benefits of iterative computation from sheer parameter scaling.
- They demonstrate the benefits of "test-time compute" scaling within a purely model-free RL context.

# Weakness
- The theoretical guarantees rely heavily on discrete assumptions—specifically binary action spaces, deterministic transitions, and string-based state representations.
- IRU is a deliberately minimal architecture meant to isolate the effect of computation. It remains an open question whether these exact compute-scaling benefits will translate easily to highly complex, high-dimensional, or pixel-based real-world robotics tasks without integrating more advanced architectures like Transformers.

---

> ### Author Rebuttal · Authors · 2026-03-30
>
> We thank the reviewer for their insightful feedback and constructive comments. We address all the reviewer’s concerns below:
>
>
> > reliance of theory on discrete action spaces and deterministic transitions & how to intuitively bridge the gap
>
> **deterministic transitions**: We would like to note that while our proof constructs a deterministic MDP, the theory is not reliant on it. Since deterministic MDPs are a special case of stochastic MDPs, and our proofs are proofs by construction, both theorems hold true for general MDPs. In fact, it is trivial to extend the constructions to stochastic transitions. We have added this extension in our paper and also updated section 4 to reflect this.
>
> **binary string representations**: We use binary string representations of states and actions to apply tools from complexity theory. For all practical purposes, this is not a restriction as any number on a computer can always be represented as a string of 0s and 1s. The underlying assumption is that numbers are represented using finite precision, which is always true practically. For example, in our experiments all states and actions are represented using 32 bits (Float32)
>
> **discrete action spaces**:  As mentioned above, in all real world experiments, continuous variables have some finite precision. For example, our experiments in OGBench use 32 bit floating point precision. These finite precision continuous variables can be viewed as finite length binary strings, hence are effectively discrete. Although our theory does not account for measure theoretical aspects of continuous variables, for all practical purposes it still holds true with finite precision "continuous" action spaces.
>
> Real numbers can be approximated using finite binary strings of increasing lengths. But we leave a proper measure theoretic treatment to future work. We have added this discussion in the main body and the limitation section of our paper.
>
>
> > detailed account of inference/training speed
>
> We looked at the average inference and training time for all the models we tried on the boxpick generalized 4-1 task. For inference, we look at a batch size of 1 and for training a batch size of 1024. The results below show that the inference time for IRU(10) is about half that of the deep ResNet. While both IRU and ResNet have similar training time. Infact, the inference time of the IRU(10), which is dominated by the python to XLA dispatch time, is comparable to that of the MLP.
>
> | Model | Single batch Inference (ms) | Train batch (ms) |
> | --- | --- | --- |
> | MLP | 0.5028 ± 0.8327 | 1.7866 ± 1.0327 |
> | ResNet | 1.3396 ± 1.1906 | 4.5833 ± 1.4296 |
> | IRU(1) | 0.2865 ± 0.4069 | 1.7795 ± 0.5779 |
> | IRU(5) | 0.3192 ± 0.1705 | 4.4713 ± 1.0699 |
> | IRU(10) | 0.6001 ± 0.5295 | 7.4494 ± 0.9058 |
> | LSTM(1) | 0.2524 ± 0.1185 | 2.5822 ± 0.8037 |
> | LSTM(5) | 0.4180 ± 0.1371 | 6.9988 ± 0.9279 |
> | LSTM(10) | 0.7089 ± 0.5510 | 12.4636 ± 0.9344 |
>
>
> > experiments with more complex architectures or robotics environments
>
> We perform some preliminary experiments with LSTMs using variable iterative steps on 2 environments – boxpick exact-4, generalized-4-1.
>
> We see that more compute does lead to improved performance; LSTM (1) always has the least performance despite using the same parameters. But the trend with more compute is not always consistent for LSTMs. We hope these results serve as a proof of concept and future work builds on finding better techniques to scale models with compute.
>
> | Iterative Steps | 1 | 2 | 5 | 10 |
> | --- | --- | --- | --- | --- |
> | Generalized 4-1 | 0.18 ± 0.04 | **0.95 ± 0.02** | 0.84 ± 0.008 | 0.85 ± 0.07 |
> | Exact | 0.0 ± 0.0 | 0.41 ± 0.009 | **0.67 ± 0.26** | 0.49 ± 0.13 |
>
> We would additionally like to point out that the main goal of our paper is to formalize compute adaptive policies and prove fundamental statements about the effects of policy compute on its performance. The MDPs used in our experiments are already high dimensional  (2^(21) states in lightsout, O(10^6) in boxpick, continuous in OGBench). While we already have results on robotics tasks (Figure 3), integrating IRUs with Transformers or testing them on pixel-based control remains an exciting avenue for future work. We have added a note for this in the future work section of our paper. **We hope that our experiments (which span over 32 diverse tasks) are a strong proof of concept that using more compute without increasing policy parameters does lead to improved performance and generalization.**
>
> **Overall, we appreciate the reviewer's acknowledgement of our contributions. Do the revisions and discussions above address the reviewer’s concerns?**

---

> > ### Author Rebuttal · Reviewer_LiRX · 2026-04-04
> >
> > Thanks for the sincere rebuttal; they resolved my concerns. I have no additional questions and will adjust the score.

---

> > > ### Author Response · Authors · 2026-04-06
> > >
> > > We are glad to hear that our rebuttal addressed your concerns. Regarding adjusting the score, we would appreciate it if you could ensure the official score on the portal reflects your updated evaluation of the work.
> > >
> > > Thank you again for your time and constructive feedback.

---

### Decision · Program_Chairs · 2026-04-30

**Decision:**

Accept (spotlight)

**Comment:**

This paper investigates the relationship between a reinforcement learning policy's computational budget and its performance, specifically addressing whether more compute can benefit a policy with a fixed number of parameters. The authors use complexity theory and time-bounded Turing machines to show that larger computational budgets enable policies to solve complex MDPs and generalize to longer-horizon tasks that are intractable for compute-constrained policies. Empirically, they introduce the Interpolation Recurrent Unit (IRU), a minimal architecture designed to isolate the effects of iterative computation from parameter scaling. Across 31 tasks, the IRU demonstrates superior performance and generalization compared to MLPs and ResNets, even when those baselines use 5 times as many parameters. Strengths include a novel and clean framing of compute as a separate axis from parameter count, a methodologically sound empirical setup, and insightful results from yoked experiments suggesting compute improves expressivity. However, weaknesses noted by reviewers include the reliance on discrete assumptions for theoretical proofs, initial non-monotonic performance results, and a lack of initial reporting on the wall-clock latency costs of additional compute steps.

The authors addressed several key concerns through their rebuttal by providing new experimental data and theoretical clarifications. To address non-monotonic performance, they performed hyperparameter tuning on learning rates and batch sizes, resulting in consistent performance improvements as the number of compute steps increased from 1 to 10. They also included a new baseline comparison using a smaller ResNet with parameter counts matched to the IRU, which further supported their claims. Regarding theoretical limitations, the authors clarified that their proofs can be extended to stochastic MDPs and argued that finite-precision continuous variables can be effectively treated as discrete binary strings. Additionally, they provided a detailed table comparing wall-clock inference and training times, showing that while IRU(10) is slower than a 1-step model, its inference latency remains significantly lower than a deep ResNet. Gradient stability was also addressed through a didactic experiment demonstrating that IRUs maintain healthy gradients over long sequences, unlike standard RNNs. No significant remaining concerns were noted, as the reviewers shifted toward "resolved" statuses following these updates.

I recommend accepting this submission because it provides a strong theoretical and empirical foundation for separating computation time from parameter scaling in reinforcement learning, a distinction often ignored in the field. The authors successfully resolved initial reviewer skepticism regarding hyperparameter tuning and theoretical assumptions by providing extensive new experiments and clarifications.